# Fused 1,2-Diboraoxazoles Based on *closo*-Decaborate Anion–Novel Members of Diboroheterocycle Class

**DOI:** 10.3390/molecules26010248

**Published:** 2021-01-05

**Authors:** Vera V. Voinova, Nikita A. Selivanov, Ivan V. Plyushchenko, Mikhail F. Vokuev, Alexander Yu. Bykov, Ilya N. Klyukin, Alexander S. Novikov, Andrey P. Zhdanov, Mikhail S. Grigoriev, Igor A. Rodin, Konstantin Yu. Zhizhin, Nikolay T. Kuznetsov

**Affiliations:** 1Kurnakov Institute of General and Inorganic Chemistry, Russian Academy of Sciences, Leninskii pr. 31, 119991 Moscow, Russia; veravoinova@rx24.ru (V.V.V.); GooVee@yandex.ru (N.A.S.); bykov@igic.ras.ru (A.Y.B.); klukinil@gmail.com (I.N.K.); zhizhin@igic.ras.ru (K.Y.Z.); ntkuz@igic.ras.ru (N.T.K.); 2Chemistry Department, Lomonosov Moscow State University, 119991 Moscow, Russia; plyush1993@bk.ru (I.V.P.); vokuevmihail11@gmail.com (M.F.V.); igorrodin@yandex.ru (I.A.R.); 3Institute of Chemistry, Saint Petersburg State University, Universitetskaya Nab. 7-9, 199034 Saint Petersburg, Russia; ja2-88@mail.ru; 4Frumkin Institute of Physical Chemistry and Electrochemistry, Russian Academy of Sciences, Leninskii pr. 31, Bldg 4, 119071 Moscow, Russia; mickgrig@mail.ru

**Keywords:** *closo*-decaborate, iminol, diboraoxazole, intramolecular ring-closure reaction, QTAIM analysis

## Abstract

The novel members of the 1,2-diboraoxazoles family have been obtained. In the present work, we have carried out the intramolecular ring-closure reaction of borylated iminols of general type [B_10_H_9_N=C(OH)R]^−^ (R = Me, Et, ^n^Pr, ^i^Pr, ^t^Bu, Ph, 4-Cl-Ph). This process is conducted in mild conditions with 83–87% yields. The solid-state structures of two salts of 1,2-diboraoxazoles were additionally investigated by X-ray crystallography. In addition, the phenomena of bonding interactions in the 1,2-diboraoxazole cycles have been theoretically studied by the Quantum Theory of Atoms in Molecules analysis. Several local and integral topological properties of the electron density involved in these interactions have been computed.

## 1. Introduction

The chemistry of boron cluster compounds and their derivatives has been extensively studied recently. This science field contains many molecule subclasses with different structure [1,2,3]. All of them have unique physicochemical behavior, which is differed from other chemical compounds. One of the distinguishing features is that boron clusters can stabilize *exo*-polyhedral substituents with unusual atomic valence states. For example, there are several examples of oxonium and nitrilium derivatives of *nido*-carboranes and *closo*-borate anions [4,5,6]. These derivatives are stable under an air atmosphere and in acid or base medium.

Boron clusters are versatile building blocks used for design compounds with given properties. Varying the structure of the boron cage and the nature of *exo*-polyhedral substituents, one can create molecules with plenty of useful properties such as high thermodynamic stability [7,8], magnetic properties [9], and low toxicity [10,11,12]. These properties cause a lot of boron cluster potential application in medicine [13,14,15], catalysis [16,17], the creation of luminescent materials [18,19], and in supramolecular chemistry [20,21].

One of the most exciting applications of boron cluster compounds is the construction of a borylated analogue of organic molecules. Encapsulation of the boron cage fragment into a molecule can have a dramatic effect on the properties of desired substances. The novel class of borylated heterocycles has been invented [22,23,24]. Such systems are based on carboranes, *nido*-carboranes and *closo*-borate anions [25]. The fused cycle of these compound contains one or two boron atoms. One of the primary approaches of borylated heterocycle formation is based on a process of metal-complex catalysis. Another one is based on using phenyliodine(III) diacetate as an inductor for the cyclization process [26]. This method has been used for the preparation of borylated diboraoxazoles of general type [B_12_H_10_ONCR^1^]^−^ (R^1^ = Aryl, Alkyl). For this process, borylated amide [B_12_H_11_NHC(R^1^)O]^−^ was chosen as the starting material. The mechanism was described [27]. Hydrolysis of the [B_12_H_10_ONCR^1^]^−^ anions leads to the preparation of derivatives of the general type [B_12_H_10_OHNH_3_]^−^

In the present work, we focused on the formation of borylated diboraoxazole systems based on *closo*-decaborate anion [B_10_H_10_]^2−^. This anion has been extensively studied recently. Similarly, for other *closo*-borate anions of general type [B_n_H_n_]^2−^ (n = 6–12), the main approaches for their functionalization are based on electrophilic and nucleophilic processes. Using these methods, derivatives with *exo*-polyhedral B–C [28,29,30], B–N [31,32,33], B–O [34,35,36], B–S [37,38,39], and B–Hal [40,41] (Hal =F,Cl,Br,I) bonds have been received. Nitrilium derivatives of general type [B_n_H_n_N≡CR^2^]^−^ (R^2^ = Me, Et, ^t^Bu, Ph) (n = 10,12) are a very suitable molecular platform for creating molecular systems with given properties [42,43,44]. These compounds contain the activated C≡N bond and can be functionalized using nucleophilic addition process. Using H_2_O as a nucleophile, one can obtain borylated iminol of the general type [B_10_H_9_N=C(OH)R^2^]^−^ [45].

Thus, the main idea of the present research is to study the possibility of the reaction of intramolecular cyclization to be carried out with the participation of iminols [B_10_H_9_N=C(OH)R]^−^ (R = Me, Et, ^n^Pr, ^i^Pr, ^t^Bu, Ph, 4-Cl-Ph), as one of the active forms in such processes.

## 2. Results and Discussion

### 2.1. Synthesis of closo-Decaborate Iminols

We have previously reported water nucleophilic addition to some nitrilium *closo*-borates [2-B_10_H_9_NCR]^1−^ (R = Me, Et, ^t^Bu, Ph), which proceeds by mild conditions giving *closo*-decaborate iminols. The iminol form is very stable both in the solutions and solid state (the crystal structures for three iminols has been reported [45,46,47]). However, under the action of bases, it could be transferred to the amide form. Meanwhile, when amides react with acids, the reversed formation of iminols proceeds (Scheme 1).

In the present study, three new iminols were obtained, which afforded expanding the range of substituents and showing the versatility of the functionalization method. The reactions were carried out at room temperature in the mixture of dichloromethane and water and completed within 6 h. The process occurs regioselectively and stereoselectively to form iminols with Z-configuration of the double bond. According to ^11^B NMR spectra, the product yields are quantitatively independent of the structure of the R-substituent.

The obtained iminols were characterized by multinuclear NMR spectroscopy and IR spectroscopy. These spectra are typical for *closo*-decaborates with iminol-substituents. The ^11^B NMR spectra of **2c, 2d, 2g** exhibit a characteristic low-field shift of the signal from the substituted boron atom in the region −16.0 to −16.7 ppm. The significant shifts of signals from apical boron atoms are associated with the presence of an intramolecular O-H ···H-B dihydrogen bond, so the signal of the B(10) atom appears in the region 3.9–2.9 ppm, and the signal of the B (1) atom appears in the region −5.7 to −6.6 ppm. The ^1^H NMR spectra of **2c, 2d, 2g** exhibit two characteristical broad singlets at δ 8.50–7.94 ppm (NH hydrogen atom) and δ 3.84–3.83 ppm (OH hydrogen atom). In the low field of the ^13^C NMR spectra, the imine carbon atom displays the singlet in region 181.6–178.4.

The solid-state structures of **2e** and **2g** were additionally investigated by X-ray crystallography (Figure 1 and Figure 2). The structure of the substituents in the obtained compounds is in good agreement with the already known examples [45,46,47]. The bond lengths in the N (1) -C (1) -O (1) iminol moiety indicate the presence of conjugation. In addition, a distinctive feature of the structures is the presence of a dihydrogen bond between the hydroxyl group and the hydride hydrogen atom of the cluster. The value of the length of the dihydrogen bond is of the order of 2.0–2.1 Å (which is significantly less than the sum of the van der Waals radii of 2.4 Å). This non-covalent interaction stabilizes the Z-configuration of the double bond and probably determines the stereoselectivity of the process.

### 2.2. Intramolecular Cyclization of Iminols

Relying on the data on the synthesis of diboraoxazoles based on the *closo*-dodecaborate anion [26], we have studied the reaction of intramolecular cyclization of iminols in the presence of hypervalent iodine compounds (Scheme 2). It has been found that 1,2-diboraoxazoles are formed as products. The reaction proceeds in aprotic solvents in an argon atmosphere with moderate heating (up to 70 °C) in 83–87% yield. Moreover, the structure of the substituent has no significant effect on the process rate or product yield. In addition, it was shown that the resulting products undergo hydrolysis in the presence of hydrazine hydrate.

The process was monitored by ^11^B NMR spectroscopy. Thus, significant changes are observed in the spectra of diboraoxazoles—the signal from the substituted apical apex undergoes a significant downfield shift and is observed in the region of 22.8–21.4 ppm, the signal from the unsubstituted apical boron atom is present in the region of −8.1 to −10.0 ppm, and the signal from the substituted equatorial boron atom falls in the range −9.5 to −11.4 ppm. Signals from unsubstituted equatorial boron atoms are observed near −26.0 to −27.6, −29.4 to −31.1, and −32.7 to −34.4 ppm.

The obtained 1,2-diboraoxazoles were fully characterized by multinuclear NMR spectroscopy, IR spectroscopy, and high-resolution electrospray ionization (ESI) mass spectrometry. In the ^1^H NMR spectra, the signal of the imine proton in the range of 9.29–8.45 ppm is characteristic for diboraoxazole rings and the signal of imine carbon atom in the ^13^C NMR spectrum is observed in the range of 193.5–186.6 ppm. In the IR spectra of **3a**–**g**, the significant changes take place as compared to the spectrum of the starting iminols. The band of the ν(O-H) vibrations disappears, while the spectra exhibit the band of the ν (B-O) vibrations at 1380 cm^−1^ and the redshift of the ν (C=N) vibrations to the region 1560–1550 cm^−1^.

The solid-state structures of **3c** and **3d** were additionally investigated by X-ray crystallography (Figure 3 and Figure 4). Analysis of the structural data of the obtained cyclic derivatives shows that the B (1) -B (2) edge is involved in the formation of the diboraoxazole ring. The parameters of *exo*-polyhedral bonds are consistent with known ones [26] and correspond to ordinary orders.

The lengths of the C-O (1.29 and 1.32 Å) and N-C (1.29 and 1.30 Å) bonds indicate an intermediate bond multiplicity between the single and double bonds, which is associated with the presence of conjugation in the O (1) -C (1) -N (1) fragment. It should be noted that the cycle is tense, as evidenced by the distortion of the boron polyhedron and a significant deviation of the angles B (1) -B (2) -N (1) and B (2) -B (1) -O (1) from the angles corresponding to the geometry of the polyhedron. In addition, anions form dimeric associates due to the formation of intermolecular dihydrogen bonds.

### 2.3. Theoretical Calculation of [B_10_H_8_ONHCR]^−^

A theoretical investigation of [B_10_H_8_ONHCR]^−^ (R = H, CH_3_, C_3_H_7_, *iso*-C_3_H_7_) has been carried out (ωB97X-D3/6-31++G(d,p) level of theory) in order to study the bonding parameters in diboraoxazole cycles. For better understanding the nature of bonds in studied compounds, we have also investigated similar contacts for related compounds. We have compared these parameters with analog for [B_12_H_11_ONHCR]^−^ (R = H, CH_3_, C_3_H_7_, *iso*-C_3_H_7_) and for [B_n_H_n−2_OHNH_3_]^−^ (n = 10, 12). The bonding interactions in the derivatives of *closo*-borate anions [B_n_H_n−2_ONHCR]^−^ have been theoretically studied using bond order analysis with the help of Wiberg bond order indices [48] and the Quantum Theory of Atoms in Molecules (QTAIM) [49], and several local and integral topological properties of the electron density involved in these interactions have been computed. We have represented all the main information for [B_10_H_8_ONHCH]^−^ and [B_n_H_n−2_OHNH_3_]^−^ (n = 10, 12) in graphical form (Figure 5). We have represented key features of these types of anion since properties of other anions are the same. For properties of all other anions, see the Appendix A.

*Bond Geometry Parameters.* We have considered main geometry parameters of the [B_10_H_8_ONHCR]^−^ (R = H, CH_3_, C_3_H_7_, *iso*-C_3_H_7_) model species (optimized structures of [B_10_H_8_ONHCH]^−^ and [B_n_H_n−2_OHNH_3_]^−^ (n = 10, 12) shown in Figure 1). The [B_10_H_8_ONHCC_3_H_7_]^−^ and [B_10_H_8_ONHC*iso*-C_3_H_7_]^−^ anions were found to be in good agreement with the experimental values described above. First, the B–O and B–N bond lengths in [B_10_H_8_ONHCR]^−^ have been examined. The B–O bond lengths lie in the interval 1.50–1.51 Å. The B–N bond lengths are 1.53–1.54 Å. In [B_n_H_n−2_OHNH_3_]^−^ (n = 10, 12) anions, the B–O bonds are shorter than in [B_n_H_n−2_ONHCR]^−^ (n = 10, 12), whereas the B–N bonds are longer than those in [B_n_H_n−2_ONHCR]^−^. The C–O bond lengths fall in the interval 1.28–1.29 Å. The C–N bond lengths are equal to 1.31 Å. For [B_12_H_11_ONHCR]^−^, the bond lengths are slightly longer than that for [B_10_H_8_ONHCR]^−^. Thus, we have found that the nature of the substituent attached to the carbonyl atom and the structure of the boron cluster have a slight impact on the bond length in the considered compounds, and all anions have similar geometry parameters.

*Wiberg bond order indices.* We have found that for all derivatives of the general form [B_10_H_8_ONHCR]^−^ (R = H, CH_3_, C_3_H_7_, *iso*-C_3_H_7_), the B–N bonds are characterized by a higher value of Wiberg index than the B–O bonds. Similarly to bond lengths, the nature of the carbonyl atom substituent has a slight impact on bond order index in considered compounds, and all anions have similar bond indices (Figure 6). The C–N bonds are characterized by a higher value of Wiberg index than the C–O bonds. For anions [B_12_H_10_ONHCR]^−^ (R = H, CH_3_, C_3_H_7_, *iso*-C_3_H_7_), all the main values of bond indexes are the same as for [B_10_H_8_ONHCR]^−^ (R = H, CH_3_, C_3_H_7_, *iso*-C_3_H_7_). In addition, we have investigated the values of Wiberg index for [B_n_H_n−2_OHNH_3_]^−^. In these anions, the B–O bonds are characterized by a higher value of bond order index than the B–N bonds.

*Topological parameters.* Similarly to the bond lengths and Wiberg indexes, we have found that the nature of the substituent attached to the carbonyl atom and the structure of the boron cluster has a slight impact on the main topological parameters of electron density in the considered compounds, and all anions have similar topological parameters (Figure 7 and Figure 8). We have found that the B–N bonds are characterized by higher values of electron density *ρ*(*r*) at bond critical points and delocalization indexes than the B–O bonds. Moreover, the B–N bonds have a more negative value of total energy as compared with the B–O bonds. The Laplacian of electron density ∇^2^*ρ*(*r*) for both types of the *exo*-polyhedral bond is positive. In case of [B_n_H_n−2_OHNH_3_]^−^, the B–O bonds are characterized by higher values of electron density *ρ*(*r*) at bond critical points and delocalization indices than the B–N bonds and have a more negative value of total energy compared with the B–N bonds. The C–N bonds are characterized by higher values of electron density *ρ*(*r*) at bond critical points and delocalization indices than the C–O bonds. In addition, the C–N bonds have a more negative value of total energy comparing with the C–O bonds. The Laplacian of electron density ∇^2^*ρ*(*r*) for the C–O and C–N bonds is negative. 

*Non-covalent interactions.* For [B_n_H_n−2_OHNH_3_]^−^ (n = 10, 12) we have found non-covalent interactions between the hydrogen atom of the ammonium group and the oxygen atom of the hydroxy-group. For [B_12_H_10_OHNH_3_]^−^, the length of hydrogen bonding is shorter, while the Wiberg index, electron density at bond critical point (bcp), and delocalization index are higher. Thus, one can conclude that for this anion, the hydrogen bond is stronger than that for [B_10_H_8_OHNH_3_]^−^.

*Atomic Charges.* We have used atoms in molecules (AIM) [49] and natural bond orbital (NBO) [50] approaches to estimate atomic charges. We have focused on atomic charges in diboraoxazole cycles. Both approaches (AIM and NBO) have the same trend, but in the case of NBO, all atoms in the cycle have more positive charges for the O- and N-atoms than those in the case of AIM. The nature of the carbonyl atom substituent has a slight impact on the atomic charges in diboraoxazole cycles. The charges on C atoms lie in the range 1.32–1.34 e for AIM and 0.52–0.73 e for NBO for [B_n_H_n−2_ONHCR]^−^ (n = 10, 12) anions. The N-atoms have more negative values than the O-atoms in most cases, whereas in case of [B_n_H_n−2_OHNH_3_]^−^, the O-atoms have more negative values than the N-atoms. For [B_10_H_8_ONHCR]^−^, boron atoms attached to oxygen atoms have atom charge values falling in the range −0.05 to −0.08 e for AIM and 0.22 e for NBO, and boron atoms attached to nitrogen atom have values of 0.28 to −0.29 e for AIM and 0.11–0.12 e for NBO. For [B_12_H_10_ONHCR]^−^, boron atoms have more positive atomic charges and boron atoms attached to oxygen atoms have values in the range of 0.18 to −0.19 e for AIM and 0.31 e for NBO. The values for boron atoms attached to nitrogen atoms fall in the range of 0.23–0.25 e for AIM and 0.12–0.13 e for NBO.

Thus, we have carried out a theoretical investigation of the bonding nature in diboraoxazole cycles in [B_n_H_n−2_ONHCR]^−^ (n = 10, 12). We have found that the nature of a substituent attached to the carbonyl atom and the structure of the boron cluster have a slight impact on the main bond parameters in the considered compounds, and all anions have similar geometry parameters. For [B_n_H_n−2_ONHCR]^−^ (n = 10, 12), the B–N bonds are characterized by stronger orbital interactions comparing to the B–O bond. The B–N bonds have higher values of the Wiberg bond order index, electron density at bond critical points, and delocalization indices than the B–O bonds. In the case of [B_n_H_n−2_OHNH_3_]^−^, the B–O bonds are characterized by stronger orbital interactions compared to the B–N bond. The C–N bonds are characterized by stronger orbital interactions compared to the C–O bond. The C–N bonds have higher values of the Wiberg index, electron density at bcp, and delocalization indexes than the C–O bonds.

## 3. Materials and Methods

Elemental analysis for carbon, hydrogen, and nitrogen was performed on a CHNS-3 FA 1108 Elemental Analyzer (Carlo Erba). The boron content was determined by the ICP-MS method on an iCAP 6300 Duo atomic emission spectrometer with inductively coupled plasma at the Scientific Research Analytical Center of the Institute for Chemical Reagents & High Purity Chemical Substances, National Research Center “Kurchatov Institute”. 

IR spectra of the compounds were recorded on an Infralum FT-08 IR Fourier spectrophotometer (NPF Lumex AP) in the region 4000–400 cm^−1^ with a resolution of 1 cm^−1^. Samples were prepared as KBr pellets. 

^1^H, ^13^C, and ^11^B[^1^H] NMR spectra of solutions of the studied substances in CD_3_CN were recorded on a Bruker MSL-300 pulsed Fourier spectrometer (Germany) at frequencies of 300.3, 75.49, and 96.32 MHz, respectively, with internal deuterium stabilization. Tetramethylsilane or boron trifluoride ether was used as the external standard, respectively.

ESI mass spectra The LC system consisted of two LC-20AD pumps (Shimadzu, Japan), and the autosampler was coupled online with an LCMS-IT-TOF mass spectrometer equipped with an electrospray ionization source (Shimadzu, Japan). The HRMS spectra were acquired in direct injection mode without a column. Mass spectra were obtained in the *m*/*z* range from 120 to 700 Da (for negative ionization mode) and 100–700 for positive mode. Other MS parameters: Detector Voltage: 1.55 kV. Nebulizing Gas: 1.50 L/min. CDL Temperature: 200.0 °C CDL. Heat Block Temperature: 200.0 °C ESI Voltage: 4.50 kV. Event Time: 300 ms. Repeat: 3. Ion Accumulation: 30 ms. Instrument tuning (mass calibration and sensitivity check) was carried out before analysis. 

The full geometry optimization of all structures has been carried out at the ωB97X-D3/6-31++G(d,p) level of theory with the help of the ORCA 4.2.1 program package [51]. The spin-restricted approximation for the structures with closed electron shells was applied. Symmetry operations were not applied during the geometry optimization procedures for all structures. The Hessian matrices were calculated numerically for all optimized structures in order to prove the location of correct minima on the potential energy surfaces (no imaginary frequencies). The natural bond orbital (NBO) method has been utilized by using NBO7 program package [50]. The topological analysis of the electron density distribution with the help of the Quantum Theory of Atoms in Molecules (QTAIM) formalism developed by Bader [49] has been performed by using the Multiwfn program (version 3.7) [52].

X-ray diffraction experiments were conducted at the Center for collective use of physical methods of research of the Frumkin Institute RAS on an automatic four-circle diffractometer with a Bruker KAPPA APEX II area detector [53] (radiation MoKα) at 20 °C. The unit cell parameters were refined on all the data [54]. The structure was solved by the direct method [55] and refined by full-matrix least-squares on *F*^2^ for all data in the anisotropic approximation for all non-hydrogen atoms [56]. H atoms of a borohydride cluster were located from a difference Fourier map and refined with isotropic temperature factors equal to 1.2 *U*_eq_(B). H atoms of the organic part of the structure were placed in geometrically calculated positions with isotropic temperature factors equal to 1.2 *U*_eq_(C) for CH_2_ groups and 1.5 *U*_eq_ (C) for the CH_3_ ones. The absolute structure was not determined. The atomic coordinates are deposited with the Cambridge Crystallographic Data Centre (Deposition Numbers 2049777-2049780). Images are created using the OLEX2 package [57].

Solvents of chemically pure grade and special quality grade (Russian State Standart) were used without further purification. (Diacetoxyiodo)benzene PhI(OAc)_2_ was used without additional purification. Nitrilium derivatives **1a**–**g** were synthesized by described methods [58].

*Synthesis of N-Borylated Iminols*: Nitrilium derivatives (Bu_4_N)[2-B_10_H_9_(NCR)] (**1a**–**g**) (0.75 mmol) were dissolved in a mixture of H_2_O (5 mL) and CH_2_Cl_2_ (5 mL). The reaction mixture was stirred under a dry argon atmosphere at room temperature for 6 h. The organic phase was dried over anhydrous sodium sulfate and concentrated on a rotary evaporator. The solid residue was recrystallized from mixture CH_3_CN/Et_2_O (1:10) and dried over P2O5. 

Spectral data for [1a-d] was reported [45].


**(Bu_4_N)[2-B_10_H_9_(NHC(OH)^n^C_3_H_7_)] 2c** Yield: 83%. ^1^H-NMR (CD_3_CN) δ (ppm): 8.06 (1H, NH), 3.84 (s, 1H, OH), 3.11 (8H, NBu_4_), 2.36 (t, 3H, *J* = 7.4 Hz, -C-CH_2_-CH_2_-CH_3_), 1.63, 1.62 (10H, NBu_4_, -C-CH_2_-CH_2_-CH_3_), 1.37 (8H, NBu_4_), 1.00 (12H, NBu_4_), 0.90 (t, 3H, J=7.4 Hz, –CH_3_)). ^11^B[^1^H]-NMR (CD_3_CN) δ (ppm): 2.2 (s, 1B, B(1)), –8.6 (s, 1B, B(10)), –18.6 (s, 1B, B(2)), –26.5, –29.7 (c.m., 7B, B(3-9)). ^13^C-NMR (CD_3_CN) δ (ppm): 178.4 (-N=C(C_3_H_7_)), 58.4 (NBu_4_), 34.7 (NH-C-CH_2_-CH_2_-CH_3_), 23.3 (NBu_4_), 19.4 (NBu_4_), 18.7 (NH-C-CH_2_-CH_2_-CH_3_), 12.8 (NBu_4_), 12.4 (NH-C-CH_2_-CH_2_-CH_3_). Anal. Calc. for C_20_H_54_B_10_N_2_O, (448.5): C, 53.77; H, 12.18; N, 6.27; B, 24.2. Found: C, 53.79; H, 12.20; N, 6.25; B, 24.2. IR (KBr, cm^−1^, selected bands): ν(OH) 3531, ν(NH) 3306, ν(BH) 2485, ν(C=N) 1646, ν(BN) 1233.

**(Bu_4_N)[2-B_10_H_9_(NHC(OH)^i^C_3_H_7_)] 2d** Yield: 82%. ^1^H-NMR (CD_3_CN) δ (ppm): 7.94 (1H, NH), 3.84 (s, 1H, OH), 3.12 (8H, NBu_4_), 2.64 (hept, 1H, *J* = 6.5 Hz, -CH-(CH_3_)_2_), 1.64 (8H, NBu_4_), 1.38 (8H, NBu_4_), 1.15 (d, 6H, *J* = 6.9 Hz, –(CH_3_)_2_), 1.00 (12H, NBu_4_). ^11^B[^1^H]-NMR (CD_3_CN) δ (ppm): 2.1 (s, 1B, B(1)), –8.6 (s, 1B, B(10)), –18.4 (s, 1B, B(2)), –26.4, –29.6 (c.m., 7B, B(3-9)). ^13^C-NMR (CD_3_CN) δ (ppm): 181.6 (-N=C(C_3_H_7_)), 58.4 (NBu_4_), 33.2 (-CH-(CH_3_)_2_), 23.4 (NBu_4_), 19.4 (NBu_4_), 17.8 (-CH-(CH_3_)_2_), 12.8 (NBu_4_). Anal. Calc. for C_20_H_54_B_10_N_2_O, (448.5): C, 53.77; H, 12.18; N, 6.27; B, 24.2. Found: C, 53.78; H, 12.18; N, 6.23; B, 24.1. IR (KBr, cm^−1^, selected bands): ν(OH) 3531, ν(NH) 3306, ν(BH) 2485, ν(C=N) 1646, ν(BN) 1213.

**(Bu_4_N)[2-B_10_H_9_(NHC(OH)C_6_H_4_Cl)] 2g** Yield: 85%. ^1^H-NMR (CD_3_CN) δ (ppm): 8.50 (1H, NH), 7.87–7.48 (ar, 4H, C-C_6_H_4_Cl), 3.83 (s, 1H, OH), 3.11 (8H, NBu_4_), 1.63 (8H, NBu_4_), 1.37 (8H, NBu_4_), 1.00 (12H, NBu_4_). ^11^B[^1^H]-NMR (CD_3_CN) δ (ppm): 2.4 (s, 1B, B(1)), °C 7.9 (s, 1B, B(10)), °C 17.6 (s, 1B, B(2)), −26.4, −29.4 (c.m., 7B, B(3-9)). ^13^C-NMR (CD_3_CN) δ (ppm): 180.0 (-N=C(C_6_H_5_Cl)), 135.8, 129.7, 129.2, 128.7 (C_6_H_5_), 58.4 (NBu_4_), 23.4 (NBu_4_), 19.4 (NBu_4_), 12.8 (NBu_4_). Anal. Calc. for C_23_H_51_B_10_ClN_2_O, (516.5): C, 53.61; H, 9.98; N, 5.44; B, 21.0. Found: C, 53.62; H, 9.96; N, 5.46; B, 21.1. IR (KBr, cm^−1^, selected bands): ν(OH) 3534, ν(NH) 3319, ν(BH) 2486, ν(C=N) 1630, ν(BN) 1206.

*Synthesis of diboraoxazoles***3a**–**g**: The mixture of compounds **2a**–**g** (0.75 mmol) and PhI(OAc)_2_ (1.125 mmol) was suspended in 10 mL of THF. The reaction mixture was heated to 90 °C and stirred under a dry argon atmosphere for 5 h. After cooling, the resulting solution was concentrated on a rotary evaporator. The resulting solid residue was recrystallized from isopropyl alcohol and dried over P_2_O_5_.

**(NBu_4_)[B_10_H_8_NHC(Me)O)] 3a** Yield: 86%. ^1^H-NMR (CD_3_CN) δ (ppm): 8.62 (1H, NH), 3.12 (8H, NBu_4_), 2.45 (s, 3H, NCCH_3_), 1.63 (8H, NBu_4_), 1.37 (8H, NBu_4_), 1.00 (2H, NBu_4_). ^11^B[^1^H]-NMR (CD_3_CN) δ(ppm): 21.9 (s, 1B, B(1)), −9.1 (s, 1B, B(10)), −10.7 (s, 1B, B(2)), −27.0, −30.5, −33.5 (c.m., 7B, B(3-9)). ^13^C-NMR (CD_3_CN) δ (ppm): 186.6 (-N=C-O-), 58.4 (NBu_4_), 23.4 (NBu_4_), 19.4 (NBu_4_), 18.5 (NH-C-CH_3_), 12.9 (NBu_4_). MS (ESI) *m*/*z*: 174.1970 (A refers to the molecular weight of [B_10_H_8_NHC(CH_3_)O], calculated for {[A]^-^} 174.1929). Anal. Calc. for C_18_H_48_B_10_N_2_O (416.7): C, 51.88; H, 11.61; N, 6.72; B, 25.9. Found C, 51.91; H, 11.66; N, 6.73; B, 25.9. IR (KBr, cm^−1^, selected bands): ν(NH) 3323, ν(CH) 2959, ν(BH) 2487, ν(C=N) 1568, ν(BN) 1485, ν(BO) 1381.

**(NBu_4_)[B_10_H_8_NHC(Et)O] 3b** Yield: 84%. ^1^H-NMR (CD_3_CN) δ (ppm): 8.52 (1H, NH), (3.11 (8H, NBu_4_), 2.76 (q, 2H, NCCH_2_CH_3_), 1.63 (8H, NBu_4_), 1.37 (m, 11H, NBu_4,_ NCCH_2_CH_3_), 1.00 (12H, NBu_4_). ^11^B[^1^H]-NMR (δ): 21.4 (s., 1B, B(1)), –9.8 (s, 1B, B(10)), –11.3 (s, 1B, B(2)), −27.6, −31.1, −34.3 (c.m., 7B, B(3-9)). ^13^C-NMR (CD_3_CN) δ (ppm): 189.5 (-N=C(C_2_H_5_)-O-), 58.4 (NBu_4_), 26.1 (NH-C-CH_2_-CH_3_), 23.4 (NBu_4_), 19.4 (NBu_4_), 12.8 (NBu_4_), 9.2 (NH-C-CH_2_-CH_3_). MS (ESI) *m/z*: 188.2121 (A refers to the molecular weight of [B_10_H_8_NHC(C_2_H_5_)O], calculated for {[A]^-^} 188.2148). Anal. Calc. for C_19_H_50_B_10_N_2_O, (430.7): C, 52.98; H, 11.70; N, 6.50; B, 25.1. Found: C, 52.95; H, 11.72; N, 6.51; B, 25.2. IR (KBr, cm^−1^, selected bands): ν(NH) 3321, ν(CH) 2961, ν(BH) 2484, ν(C=N) 1558, ν(BN) 1486, ν(BO) 1379.

**(NBu_4_)[B_10_H_8_NHC(^n^Pr)O] 3c** Yield 85%. ^1^H-NMR (CD_3_CN) δ (ppm): 8.56 (1H, NH), 3.12 (8H, NBu_4_), 2.71 (t, 2H, *J* = 7.3 Hz, NCCH_2_CH_2_CH_3_), 1.84 (h, 2H, *J* = 7.4 Hz, NCCH_2_CH_2_CH_3_), 1.63 (8H, NBu_4_), 1.37 (8H, NBu_4_), 1.00 (m, 15H, NBu_4,_ NCCH_2_CH_2_CH_3_). ^11^B[^1^H]-NMR (CD_3_CN) δ (ppm): 21.9 (s, 1B, B(1)), −9.2 (s, 1B, B(10)), –10.9 (s, 1B, B(2)), −27.0, −30.5, −33.6 (c.m., 7B, B(3-9)). ^13^C-NMR (CD_3_CN) δ (ppm): 189.7 (-N=C(C_3_H_7_)-O-), 58.4 (NBu_4_), 34.2 (NH-C-CH_2_-CH_2_-CH_3_), 23.4 (NBu_4_), 19.4 (NBu_4_), 19.2 (NH-C-CH_2_-CH_2_-CH_3_), 12.9 (NBu_4_), 12.6 (NH-C-CH_2_-CH_2_-CH_3_). MS (ESI) *m*/*z*: 202.2268 (A refers to the molecular weight of [B_10_H_8_NHC(C_3_H_7_)O], calculated for {[A]^-^} 202.2235). Anal. Calc. for C_20_H_52_B_10_N_2_O, (444.7): C, 54.01; H, 11.78; N, 6.30; B, 24.3. Found: C, 54.02; H, 11.74; N, 6.28; B, 24.3. IR (KBr, cm^−1^, selected bands): ν(NH) 3321, ν(CH) 2964, ν(BH) 2484, ν(C=N) 1563, ν(BN) 1472, ν(BO) 1381.

**(NBu_4_)[B_10_H_8_NHC(^i^Pr)O] 3d** Yield: 83%. ^1^H-NMR (CD_3_CN) δ (ppm): 8.51 (1H, NH), 3.11 (8H, NBu_4_), 1.63 (8H, NBu_4_), 1.36 (m, 14H, NBu_4_, NC(CH_3_)_2_), 1.00 (12H, NBu_4_). ^11^B[^1^H]-NMR (δ): 21.9 (s, 1B, B(1)), −9.2 (s, 1B, B(10)), −10.9 (s, 1B, B(2)), −27.0, −30.4, −33.5 (c.m., 7B, B(3-9)). ^13^C-NMR (CD_3_CN) δ (ppm): 193.5 (-N=C(C_3_H_7_)-O-), 58.4 (NBu_4_), 32.5 (-CH-(CH_3_)_2_), 23.4 (NBu_4_), 19.4 (NBu_4_), 18.6 (C-CH-(CH_3_)_2_), 12.8 (NBu_4_). MS (ESI) *m*/*z*: 202.2269 (A refers to the molecular weight of [B_10_H_8_NHC(C_3_H_7_)O], calculated for {[A]^-^}202.2235). Anal. Calc. for C_20_H_52_B_10_N_2_O, (444.7): C, 54.01; H, 11.78; N, 6.30; B, 24.3. Found: C, 54.04; H, 11.81; N, 6.28; B, 24.3. IR (KBr, cm^−1^, selected bands): ν(NH) 3322, ν(CH) 2962, ν(BH) 2481, ν(C=N) 1560, ν(BN) 1470, ν(BO) 1381.

**(NBu_4_)[B_10_H_8_NHC(^t^Bu)O] 3e** Yield: 84%. ^1^H-NMR (CD_3_CN) δ (ppm): 8.45 (1H, NH), 3.12 (8H, NBu_4_), 1.63 (8H, NBu_4_), 1.39 (m, 9H, C(CH_3_)_3_), 1.38 (8H, NBu_4_), 1.00 (12H, NBu_4_). ^11^B[^1^H]-NMR (CD_3_CN) δ (ppm): 21.8 (s, 1B, B(1)), –10.0 (s, 1B, B(10)), −11.4 (s, 1B, B(2)), −27.7, −31.3, −34.4 (c.m., 7B, B(3-9)). ^13^C-NMR (CD_3_CN) δ (ppm): 195.3 (-N=C(C_4_H_9_)-O-), 58.4 (NBu_4_), 37.3 (C-(CH_3_)_3_), 26.5 (C-(CH_3_)_3_), 23.6 (NBu_4_), 19.4 (NBu_4_), 12.8 (NBu_4_). HR-MS ESI *m*/*z*: 217.2355 (A refers to the molecular weight of [B_10_H_8_NHC(C_4_H_9_)O], calculated for {[A]^-^}217.2365). Anal. Calc. for C_21_H_54_B_10_N_2_O, (458.8): C, 54.98; H, 11.86; N, 6.11; B, 23.6. Found: C, 54.93; H, 11.84; N, 6.14; B, 23.5. IR (KBr, cm^−1^, selected bands): ν(NH) 3286, ν(CH) 2964, ν(BH) 2478, ν(C=N) 1552, ν(BN) 1473, ν(BO) 1382.

**(NBu_4_)[B_10_H_8_NHC(Ph)O] 3f** Yield: 87%. ^1^H-NMR (CD_3_CN) δ (ppm): 9.24 (1H, NH), 8.14, 7.74, 7.64, 7.50 (ar, 5H, C-C_6_H_5_), 3.11 (8H, NBu_4_), 1.63 (8H, NBu_4_), 1.39 (8H, NBu_4_), 1.00 (12H, NBu_4_). ^11^B[^1^H] NMR (δ): 22.8 (c.m., 1B, B(1)), −8.1 (s, 1B, B(10)), −9.5 (c.m., 1B, B(2)), −26.0, −29.4, −32.7 (c.m., 7B, B(3-9)). ^13^C-NMR (CD_3_CN) δ (ppm): 173.1 (-N=C(C_6_H_5_)-O-), 133.7, 129.3, 128.1, 127.7 (C_6_H_5_), 58.4 (NBu_4_), 23.3 (NBu_4_), 19.4 (NBu_4_), 12.8 (NBu_4_). MS (ESI) *m*/*z*: 237.2039 (A refers to the molecular weight of [B_10_H_8_NHC(C_6_H_5_)O], calculated for {[A]^-^} 237.2054). Anal. Calc. for C_23_H_50_B_10_N_2_O, (478.8): C, 57.70; H, 10.53; N, 5.85; B, 22.6. Found: C, 57.68; H, 10.49; N, 5.86; B, 22.5. IR (KBr, cm^−1^, selected bands): ν(NH) 3327, ν(CH) 2964, ν(BH) 2484, ν(C=N) 1547, ν(BN) 1465, ν(BO) 1381.

**(NBu_4_)[B_10_H_8_NHC(C_6_H_4_Cl)O] 3g** Yield 83%. ^1^H-NMR (CD_3_CN) δ (ppm): 9.29 (1H, NH), 8.10, 7.63 (ar, 4H, NCC_6_H_5_), 3.11 (8H, NBu_4_), 1.63 (8H, NBu_4_), 1.39 (8H, NBu_4_), 1.00 (12H, NBu_4_). ^11^B[^1^H]-NMR (CD_3_CN) δ (ppm): 22.0 (s, 1B, B(1)), −8.8 (s, 1B, B(10)), –10.3 (s, 1B, B(2)), −26.8, −30.2, −33.5 (c.m., 7B, B(3-9)). ^13^C-NMR (CD_3_CN) δ (ppm): 180.6 (-N=C–O)), 133.0, 129.8, 129.4, 129.2 (C_6_H_4_Cl), 58.3 (NBu_4_), 23.4 (NBu_4_), 19.4 (NBu_4_), 12.8 (NBu_4_). MS (ESI) *m*/*z*: 271.1727 (A refers to the molecular weight of [B_10_H_8_NHC(C_6_H_5_Cl)O], calculated for {[A]^-^}271.1767). Anal. Calc. for C_23_H_49_B_10_ClN_2_O, (513.2): C, 53.83; H, 9.62; N, 5.46; B, 21.1. Found: C, 53.81; H, 9.63; N, 5.46; B, 21.0. IR (KBr, cm^−1^, selected bands): ν(NH) 3334, ν(CH) 2962, ν(BH) 2484, ν(C=N) 1598, ν(BN) 1471, ν(BO) 1380.

**(NBu_4_)[B_10_H_8_(OH)(NH_3_)] 4** N_2_H_4_*H_2_O (3 mL) was added to the solution of **3a**–**g** (0.50 mmol) in EtOH (5 mL). The reaction mixture was heated to reflux and stirred for 6 h. After cooling, the resulting solution was neutralized with 1M HCl to pH = 7, and the product was extracted with CH_2_Cl_2_ (2*10 mL). The organic layer was dried over Na_2_SO_4_ and concentrated on a rotary evaporator. The resulting solid residue was recrystallized from isopropyl alcohol and dried over P_2_O_5_.

Yield: 74%. ^1^H-NMR (CD_3_CN) δ (ppm): 5.18 (3H, NH), 3.68 (1H, OH), 3.12 (8H, NBu_4_), 1.63 (8H, NBu_4_), 1.37 (8H, NBu_4_), 1.00 (2H, NBu_4_). ^11^B[^1^H]-NMR (CD_3_CN) δ(ppm): 14.3 (s, 1B, B(1)), −7.5 (s, 1B, B(10)), –18.4 (s, 1B, B(2)), –28.0, –31.3 (c.m., 7B, B(3-9). MS (ESI) *m*/*z*: 151.2277 (A refers to the molecular weight of [B_10_H_8_(OH)(NH_3_)], calculated for {[A]^-^} 151.2282). Anal. Calc. for C_16_H_48_B_10_N_2_O (394.5): C, 48.67; H, 12.26; N, 7.10; B, 28.0. Found: C, 48.62; H, 12.24 N, 7.12; B, 28.1. IR (KBr, cm^−1^, selected bands): υ(NH) 3338, ν(BH) 2476, ν(BN) 1482, ν(BO) 1382.

## 4. Conclusions

The reaction of borylated iminols of the general type [B_10_H_9_N=C(OH)R]^−^ (R = Me, Et, ^n^Pr, ^i^Pr, ^t^Bu, Ph, 4-Cl-Ph) with iodine (III) reagents leads to the formation of fused 1,2-diboraoxazole-*closo*-decaborate. We have shown that iminols are a reactive form in the intramolecular cycling reaction. The parameters of diboroheterocycles have been analyzed using quantum chemical calculations.

## Data Availability

Date of the compounds are available from the authors.

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
