# Peer review of "Fused 1,2-Diboraoxazoles Based on closo-Decaborate Anion–Novel Members of Diboroheterocycle Class"

_molecules, 2021, doi:10.3390/molecules26010248_

Round 1
Reviewer 1 Report
The authors report the synthesis of new fused 1,2-diboraoxazoles via addition of water to the nitrilium compounds. Several of the compounds are characterised by single crystal X-ray diffraction. The manuscript is well prepared and I am recommending it for publication. A few small suggestions to the authors follow:
-please check through the document and keep the spelling consistent for "diboraoxazoles"
-please amend the chemical formula information in .cif for compound 2g
-in the crystallographic data for 3d, I think the model could be improved by modelling some disorder in the ammonium ion.
Author Response
Thanks a lot for the peer-review of our paper.
We have taken into account all your comments and uploaded revised files to the system.
Reviewer 2 Report
This paper reports on the preparation of 1,2-diboraoxazoles obtained by oxidative ring closure of the corresponding iminols. These compounds have been fully characterized by spectroscopic methods and have been subjected to theoretical calculations to evaluate the nature of bonding interactions. Cleavage of the oxazole ring has been pursued using hydrazine hydrate leading to the corresponding amino alcohols. Due to the anionic character of these compounds I wonder if the authors have planned a possible utilization as basic systems to catalyse or promote organic processes.
Overall, the chemistry disclosed in this paper is interesting enough to warrant publication in Molecules essentially as it stands.
Author Response
Thanks a lot for the peer-review of our paper. We have planned to investigate the coordination chemistry of obtained amino alcohols and their possible use as a radical trap.
We will definitely try to investigate these substances in organic catalytic or promoting systems.